# A Metastatic Cancer Expression Generator (MetGen): A Generative Contrastive Learning Framework for Metastatic Cancer Generation

**DOI:** 10.3390/cancers16091653

**Published:** 2024-04-25

**Authors:** Zhentao Liu, Yu-Chiao Chiu, Yidong Chen, Yufei Huang

**Affiliations:** 1Department of Electrical and Computer, University of Pittsburgh, Pittsburgh, PA 15260, USA; zhl169@pitt.edu; 2Cancer Virology Program, UPMC Hillman Cancer Center, Pittsburgh, PA 15232, USA; 3Cancer Therapeutics Program, UPMC Hillman Cancer Center, Pittsburgh, PA 15232, USA; yuc250@pitt.edu; 4Department of Medicine, University of Pittsburgh School of Medicine, Pittsburgh, PA 15213, USA; 5Greehey Children Cancer Research Institute, The University of Texas Health Science Center at San Antonio, San Antonio, TX 78229, USA; 6Department of Population Health Science, The University of Texas Health Science Center at San Antonio, San Antonio, TX 78229, USA

**Keywords:** metastatic cancer, deep learning, contrastive learning, tumor microenvironment

## Abstract

**Simple Summary:**

Metastasis, the spread of cancer cells to other parts of the body, is the leading cause of cancer-related deaths despite medical advances. The current methods used to study metastatic cancer face challenges in gathering enough data. To tackle this issue, we developed MetGen, a deep learning model that generates metastatic gene expression files using cancer and tissue samples. Our results show that the proposed model generated samples comparable to real data. The interpretability of the model could help researchers better understand cancer metastasis and lead to the discovery of new treatments to combat metastatic cancer.

**Abstract:**

Despite significant advances in tumor biology and clinical therapeutics, metastasis remains the primary cause of cancer-related deaths. While RNA-seq technology has been used extensively to study metastatic cancer characteristics, challenges persist in acquiring adequate transcriptomic data. To overcome this challenge, we propose MetGen, a generative contrastive learning tool based on a deep learning model. MetGen generates synthetic metastatic cancer expression profiles using primary cancer and normal tissue expression data. Our results demonstrate that MetGen generates comparable samples to actual metastatic cancer samples, and the cancer and tissue classification yields performance rates of 99.8 ± 0.2% and 95.0 ± 2.3%, respectively. A benchmark analysis suggests that the proposed model outperforms traditional generative models such as the variational autoencoder. In metastatic subtype classification, our generated samples show 97.6% predicting power compared to true metastatic samples. Additionally, we demonstrate MetGen’s interpretability using metastatic prostate cancer and metastatic breast cancer. MetGen has learned highly relevant signatures in cancer, tissue, and tumor microenvironments, such as immune responses and the metastasis process, which can potentially foster a more comprehensive understanding of metastatic cancer biology. The development of MetGen represents a significant step toward the study of metastatic cancer biology by providing a generative model that identifies candidate therapeutic targets for the treatment of metastatic cancer.

## 1. Introduction

Tumor metastasis refers to the process where cancer cells migrate from the primary tumor site to establish secondary growths in distant organs. As these cancer cells disseminate from the initial tumor, they typically travel through the bloodstream or lymphatic system to reach other areas of the body. Remarkably, metastatic progression accounts for over 90% of cancer-related deaths [1]. Managing metastatic cancer poses significant challenges. Patients with metastatic tumors frequently exhibit limited responsiveness to treatments that are effective against the corresponding primary cancers. This resistance can stem from the distinct biology of metastatic tumors and the absence of therapies tailored to address their unique characteristics. The genetic makeup of metastatic tumors often confers heightened resistance to standard treatments, which is exacerbated by the variability in tumor microenvironments (TMEs) that can lead to diverse responses to therapy [2]. It is necessary to characterize the complicated molecular mechanism in order to choose appropriate treatment strategies.

Recently, RNA-seq transcriptome data have been widely used to study the nature of cancer metastasis [3,4,5]. High-throughput technology frees the potential exploration of transcriptome changes at the whole-genome level, which would be used for detecting biomarkers for diagnosis and treatment.

### 1.1. Challenges in the Current State

One critical challenge in studying metastatic cancer is the difficulty of collecting data due to the complicated characteristics of metastatic cancer. Metastatic cancer is often in a quite late stage; therefore, patients’ survival rates are significantly lower than those of primary cancer patients, causing there to be much fewer participants in any clinical trial [6]. The diagnosis of metastatic cancer is also relatively difficult since the primary site and biopsy site are separated, which could be confusing at some points. Therefore, building an atlas for metastatic cancer could be costly. Even some big projects, such as The Cancer Genome Atlas (TCGA) or Integrative Clinical Genomics of Metastatic Cancer (MET500), only store a few hundred metastatic samples, including the tumor types for which the primary tumor is rarely diagnosed, such as melanoma [7,8]. It leaves great potential for metastatic cancer data augmentation.

Generative models, such as Generative Adversarial Networks (GANs), have achieved remarkable success in various fields, including computer vision and natural language processing [9]. However, despite their potential, GANs suffer from several major challenges: (1) Their training can be unstable due to the simultaneous training of the generator and discriminator models in an adversarial game, where improvements to one model come at the expense of the other, resulting in difficulty in achieving convergence. (2) GANs are prone to mode collapse, where the generator model can learn to produce samples within a particular mode, resulting in a lack of diversity in the generated data. (3) GANs require large amounts of data to produce good results. (4) The generators do not have encouragement to provide meaningful representation, and therefore have poor interpretability. In the context of metastatic data, GANs face additional challenges due to the limited availability of samples, especially for rare metastatic cancers where only a few samples are available, resulting in severe unstable training and mode collapse. Furthermore, biological data are complex and carry rich entangled information, requiring an interpretable model for downstream analysis.

### 1.2. Our Contributions

To this end, we proposed a metastatic cancer expression generator (MetGen), a contrastive learning-based generative model that can generate metastatic expression profiles from primary cancer and normal tissue expression samples. Generating metastatic samples from primary cancer and tissue samples is a challenging task mainly due to limited metastatic cancer samples and corresponding primary cancer and tissue samples. For example, the MET500 dataset has a total of 868 samples from 22 different metastatic cancer types, where most of the cancer types have less than 30 samples [8]. If we further divide one metastatic cancer into subtypes based on tissue sites, the sample size of each subtype would be even smaller, which would cause mode collapse and poor generating performance. To resolve this issue, contrastive learning is adopted in our model. The goal of contrastive learning is to learn such an embedding space in which similar sample pairs stay close to each other, while dissimilar ones are far apart [10]. Contrastive learning can be applied to both supervised and unsupervised settings. In the context of our study, our objective is to measure the similarity between a generated sample and an actual metastatic sample. Specifically, we aim to minimize the distance between samples that originate from the same type of metastatic cancer while maximizing the distance between samples from different types of metastatic cancers.

#### 1.2.1. MetGen Avoids Mode Collapse

Mode collapse is caused by the model finding a trivial solution that only fits the majority of single mode samples but ignores the samples in other modes [11]. In our model, modes are not only regulated by MET500 target samples but also TCGA source samples. All of the modes, namely metastatic cancer types in our scenario, are equally predefined and learned. Therefore, there is no risk of the model ignoring any of them.

#### 1.2.2. MetGen Ensures Stable Training

In contrastive learning, samples are fed to the model in pairs [10]. By pairing cancer, tissue, and metastatic samples recursively, we could easily obtain a massive training dataset. For example, one breast cancer sample can pair with multiple different liver tissue samples, which can result in multiple sample pairs for training. In this way, even rare metastatic cancer types could be trained to be stable and sufficient.

#### 1.2.3. MetGen Has an Interpretable Latent Space

MetGen has many advantages in model interpretation. Firstly, MetGen model’s latent code is obtained using a variational autoencoder, and to a large degree, the learned code components are disentangled [12,13]. Each component or component cluster represents different functions. Secondly, the autoencoder gives a convenient connection between a latent space and gene expression space. The changes in the latent space could be reflected in the gene expression space by reconstructing the latent code. Therefore, the biological function of each code component could easily be learned in an expression space by masking them out in a code space.

In summary, our proposed model integrates primary cancer information into the context of metastasized tissue sites during the generative process. This allows for differential analysis and interpretable visualization in a latent space. We anticipate that our model can facilitate the identification of functional pathways associated with cancer metastasis and contribute to the investigation of their roles in this process.

## 2. Materials and Methods

### 2.1. Data Preprocessing

The MET500 dataset and TCGA used to train MetGen were downloaded from UCSC XenaHub. For the sake of discussion, we use the term ‘cancer type’ for TCGA primary cancer and metastatic cancer types; ‘tissue type’ for TCGA normal tissue and metastatic biopsy sites; and ‘metastatic subtype’ for specific primary cancer migrated in specific tissue sites. Both TCGA and MET500 have a huge number of genes, and most of those genes have very low expression and discriminative power; thus, they induce noisy backgrounds, which essentially hurt the model training. Also, the two datasets have different labeling. Since the goal was to generate MET500 samples from TCGA samples, we needed to match two datasets in genes and labels.

To unify TCGA and MET500 genes, we first filtered out genes in MET500 with mean and standard deviation values of less than −5 and 1.5, respectively. A total of 6032 genes were kept after filtering. Similarly, we used the mean and standard deviation to filter TCGA data, and 4096 genes were kept. We took the union of two gene sets and then removed the genes which do not exist in both TCGA and MET500. Eventually, 7312 genes were kept for this study. Expressions in all samples were normalized and transformed into log2(FPKM + 1) and then scaled between 0 and 1 using the min–max scaler.

To unify TCGA and MET500 labels, we first combined similar cancer types in TCGA and then renamed all of the TCGA cancer labels to the corresponding cancer labels in MET500. Specifically, four primary cancer groups in TCGA, known as (COAD, READ), (KIRP, KIRC, KICH), (LUAD, LUSC), and LIHC, were renamed as their corresponding metastatic labels in MET500 [COLO, KDNY, LUNG, HCC], respectively. After relabeling, TCGA and MET500 had 20 common cancer types in total. We picked the top 6 common cancer types in MET500 as the cancer types in this study since we wanted to obtain as much target information as possible. The corresponding tissue types in both TCGA and MET500 dropped to 6 after cancer type filtering; we kept all of them as tissue types. Eventually, 3052 TCGA cancer samples in 6 cancer types, 338 TCGA tissue samples in 6 tissue types, and 219 MET500 samples in 19 metastatic subtypes were kept (Table 1).

#### Training and Testing Data Preparation

The training data have 6 cancer types and 6 tissue types in total in both TCGA samples and MET500 samples. We obtained 219 samples in 19 metastatic subtypes, such as BRCA in the bladder, SARC in the lungs, etc. For each subtype, we sampled pairs from TCGA cancer, normal tissue, and MET500 samples. If TCGA cancer plus TCGA normal tissue matched with the MET500 sample, we called it a positive pair; otherwise, we called it a negative pair. For example, TCGA breast cancer, TCGA liver tissue, and MET500 breast cancer in liver are positive pairings, while TCGA breast cancer, TCGA lung tissue, and MET500 breast cancer in bladder are negative pairings. Negative pairs are critical for contrastive learning, as increasing the number of negative pairs improves the downstream task performance [14]. We generated 1000 positive pairs and 3000 negative pairs for each subtype in a total of 76,000 sample pairs and split them into training data and validation data. We also created 3800 sample pairs, 200 pairs for each metastatic subtype, as the testing data. The testing samples were carefully selected so none of the cancer sample and tissue sample pairs were exposed to the model in training.

### 2.2. MetVAE Model

In this section, we explain the variational autoencoder used for MET500 code extraction in detail. Unlike a classic autoencoder, a variational autoencoder does not only learn the representative code, but also the prior distribution of data [13]. We modeled MET500 gene expression $x_{MET}$ from a Gaussian distribution $p\left(x_{MET}|z\right)$. Vanilla VAE encourages posterior distribution over the $p\left(x_{MET}|z\right)$ to be isotropic Gaussian $p\left(x_{MET}|z\right)$; thus, the latent code z could have disentangled features, which is convenient for further interpretation work.

Our VAE network has a standard architecture that consists of an encoder, a loss function, and a decoder (Appendix A). The number of neurons of two hidden layers in the encoder are 1000, and 100. The decoder exhibits a symmetric structure mirroring that of the encoder. All neural networks use a batch normalization layer (Table 2 and Table 3).

The goal of inference is to estimate the maximum likelihood of $p\left(x_{MET}|z\right)$, which is naturally defined using a decoder. We cannot apply the Bayes theorem directly because in general, it is an intractable computation. However, we could use the variational inference technique to low-bound the likelihood, which can be expressed as
(1)$$logp\left(x_{MET}|z\right)\geq E_{q\left(z|x_{MET}\right)}logp\left(x_{MET}|z\right)-D_{KL}(q(z|x_{MET})||p\left(z\right))$$
where q(z|x) is predefined to be Gaussian with a diagonal covariance matrix. The mean and covariance are learned from the encoder network.

We use the Adam optimizer to train the model with an initial learning rate of 0.0001 and a decay rate of 1 × 10^−6^. The model is trained in a batch size of 32 and optimizes the objective function. When the loss is stable for more than 20 epochs, we stop the training since the model is converged. In our case, it usually takes around 200 epochs. After the model is well trained, we save code *z* of MET500 for further studies.

### 2.3. Contrastive Learning for MetGen

In this section, we explain the MetGen network in detail. The whole framework consists of three modules, namely TCGA encoder, the CNN mixer, and the contrastive learner (Appendix A).

#### 2.3.1. TCGA Encoder

The performance of machine learning algorithms can be dramatically impacted by too many features, which is a phenomenon generally referred to as a ‘curse of dimensionality’. The reason for the curse of dimensionality is essentially caused by the complication of a large data space, which contains noise and unnecessary information. When dealing with high dimensional data, it is often efficient to reduce the number of features by projecting the data to a lower dimensional space:(2)$$code\left(x_{cancer},x_{tissue};\theta \right)=E\left(x_{cancer,}x_{tissue};\theta \right)$$
where $x_{cancer}$ and $x_{tissue}$ are input TCGA samples, E is the encoder, and θ represents the parameters of the encoder.

To capture the features of TCGA data, we adopted the architecture from a published work [15]; it was built using two CNN layers and one fully connected layer. The previous work used all 33 cancer types and normal tissue for training, and since our data are relatively small, we reduced the number of parameters of the previous model to obtain a smaller model space (Table 4).

#### 2.3.2. CNN Mixer

A CNN mixer is the model merging cancer code and tissue code used to generate a metastatic cancer code. It learns the mechanism of tissue environment activation for cancer invasion.

A CNN mixer consists of two CNN layers and one fully connected layer. First, the module stacks the cancer code and tissue code to the 2 × 512 matrix, and then it feeds the matrix to the first CNN layer. We used 64 kernels in a 2 × 1 matrix to learn the merging process. Since the cancer code and tissue code are in the same space, the 2 × 1 kernel gives us better biological meaning, which is helpful in interpretation (Table 5). The second CNN layer and fully connected layer are used to project the output code to a dimension of 100 so that we can compare it with the corresponding MET500 code.
(3)$$code\left(x_{MET_{mix}};\omega \right)=f\left(E\left(x_{cancer}\right),E\left(x_{tissue}\right);\omega \right)$$
where *f* is the CNN mixer module, and *ω* represents the parameters of the CNN mixer.

#### 2.3.3. Contrastive Learner

Contrastive loss takes the output of a positive example and calculates its distance to an example in the same class and contrasts it with the distance of negative examples [10]. In our case, the network will calculate the Euclidian distance of the generated code and MET500 code. When the labels of two codes match, the Euclidian distance should be minimized to zero; otherwise, the distance would be pushed to a margin. Thus, our final loss function is
(4)$$\begin{array}{l} L_{contrastive}\left(x_{cancer},x_{tissue},z;\theta ,\omega \right)\\ \;\;\;\;\;\;\;\;\;\;\;\;\;\;\;\;\;\;\;\;\;\;=\;y_{true}\times ED^{2}\left(code\left(x_{MET_{mix}};\omega \right),z\right)\\ \;\;\;\;\;\;\;\;\;\;\;\;\;\;\;\;\;\;\;\;\;\;+\left(1-y_{true}\right)\times \max\left(margin-ED\left(code\left(x_{MET};\omega \right),z\right),0\right) \end{array}$$
where $ED\left(,\right)$ is the Euclidian distance, $y_{true}$ is the label of the sample pair, and 1 and 0 represent positive and negative, respectively.

### 2.4. Overview of MetGen Framework

The MetGen framework consists of two model components, MetVAE and MetGen. Establishing the MetGen framework includes three stages, i.e., (1) training MetVAE to obtain latent metastatic cancer codes for MET500 samples, (2) training MetGen to generate metastatic cancer codes through contrastive mixing of cancer and tissue expressions, and (3) reconstructing metastatic cancer expression from the generated metastatic cancer code (Figure 1).

In the first stage, MetVAE was trained using gene expression profiles from the Integrative Clinical Genomics of Metastatic Cancer (MET500). MetVAE is a variational autoencoder (VAE) that includes an encoder and a decoder (Figure 1a). The MetVAE encoder converts a metastatic gene expression sample to a latent metastatic code distribution, which represents a low dimensional representation of the input expression. MetVAE decoder takes a sample from the latent code distribution and reconstructs the input gene expression. MetVAE was trained to minimize the VAE loss that enforces the reconstructed sample to be close to the input while minimizing the KL divergence between the learned latent code distribution and standard Gaussian distribution. Therefore, MetVAE learned to produce good latent representations of input MET500 metastatic samples.

In the second stage, MetGen was trained using TCGA primary cancer and tissue expression samples to generate the latent metastatic code of a metastatic cancer type from the corresponding paired primary cancer and tissue expression data (Figure 1b). MetVAE includes two expression encoders with shared weights followed by a CNN mixer (Figure 1b). The two expression encoders convert TCGA primary cancer and tissue gene expression data to primary cancer and tissue latent codes, respectively. The two encoders, which are shared-weight convolutional neural networks (CNNs), extract cancer and tissue features and filter useless information to generate metastatic cancer samples. The CNN mixer integrates cancer and tissue codes to generate the corresponding metastatic cancer code, i.e., the code for the corresponding input primary cancer spreading to the input tissue. At the input of the CNN mixer are stacked cancer and tissue codes (Figure 1b), which are compressed into a 1D code by multiple 2 × 1 kennels, each mixing them into different metastatic cancer-related features. Two fully connected layers are followed to produce nonlinear combinations of these features and generate the metastatic cancer code. The expression encoder and CNN mixer are trained with a contrastive learning loss to make the generated codes follow the distribution of MetVAE codes of the corresponding metastatic cancer in MET500. The contrastive learning loss compares the generated code and a MET500 metastatic code and minimizes the distances between the codes of matching metastatic cancer types (e.g., when the MetGen output code is generated from TCGA breast cancer and live tissue and MetVAE code is from MET500 breast cancer in the liver) but keeps the distance of codes from mismatched types at a preset margin. When the training converges, MetGen generates metastatic cancer codes as close to the MET500 codes of the matching metastatic cancer type as possible, but they are not to be confused with other metastatic cancer types.

In the third stage, the MetGen-generated metastatic code is fed into the MetVAE decoder to reconstruct the corresponding metastatic cancer gene expression profile (Figure 1c).

### 2.5. Statistical Tools

#### 2.5.1. Standard DNN Classifiers

The classifier used in all of the classification tasks adopted the same architecture. We constructed a two-layer MLP neural network, which has 60 and 40 neurons in each layer. The activation function was chosen as ReLU. Then, the model feeds extracted low dimensional features to SoftMax classifier for prediction (Table 6).

All of the deep learning models were trained on Keras version 2.4 API with TensorFlow backend [16].

#### 2.5.2. GSVA

A gene set variation analysis (GSVA) is a nonparametric unsupervised analysis method mainly used to evaluate the gene set enrichment of single samples [17]. The expression matrix of genes is transformed into the expression matrix of pathways in different samples to evaluate different pathways enriched in different samples. It stabilizes the gene expression signals in robust high-level functions. In this study, we performed GSVA using the GSVA package version 1.36.3 in R version 4.0 software. KEGG and HALLMARK gene sets were downloaded from MSigDB collection [18].

#### 2.5.3. Differential Analysis

The Bioconductor Linear Model for Microarray Analysis (LIMMA) package version 3.46.0 was used to calculate the differential pathway expression of each cancer sample in the present study. LIMMA remains highly recommended for such analyses since the pathway expressions from GSVA are continuous numbers, which is more similar to microarray analysis [19].

## 3. Results and Discussion

### 3.1. MetVAE Codes Capture Cancer and Tissue Features of Metastatic Samples

We first examined whether MetVAE codes could represent metastatic samples in lower dimensions without compromising the feature patterns. The expressions and MetVAE latent codes were visualized using t-SNE [20] (Figure 2). In the expression space, unlike primary tumors [21] (Appendix A), the metastatic samples did not form distinguished clusters, though most of the cancer types were loosely grouped, such as BRCA and PRAD (Figure 2a). On the other hand, the tissue types were not segregated based on biopsy sites, with the exception of the liver [8] (Figure 2b). In the code space, the cancer types showed a similar pattern to samples in the gene expression space (Figure 2c); however, the cancer groups were more clustered in the code space than in the gene expression space. For example, BRCA, HNSC, and PRAD were further separated from other cancer groups and formed clear clusters. For the t-SNE tissue types (Figure 2d), no significant change was discovered. We still observed liver tissues clustered together, while other tissue types were loosely distributed. In general, the MET500 samples grouped in cancer types, and within cancer clusters, we could observe multiple tissue sites. This was expected because metastatic cancer is the primary tumor cell that is metastasized to other tissue sites. They are still considered to be the same cancer type as primary tumors, so their characteristics are similar overall. In summary, our MetVAE model learned faithful patterns of the cancer types, tissue types, and metastatic subtypes in the code space compared to the gene expression space if not better.

To further quantify the extent to which the latent metastatic cancer codes captured both cancer and tissue features, we trained two classifiers on MET500 codes to predict cancer types and tissue types, respectively. The cancer classifier and tissue classifier achieved 98.3 ± 0.7% and 92.3 ± 0.6% ROC AUCs, indicating that MetVAE could correctly capture both the cancer and tissue information of the MET500 samples.

### 3.2. MetGen Generates High-Quality Metastatic Cancer Samples

We generated 200 samples for every 19 metastatic subtypes, resulting in 3800 samples, using testing data. To visualize the results, t-SNE was applied to reduce the feature dimensions. We observed 19 metastatic subtype clusters, while the original MET500 data did not form such clear cancer and tissue clusters (Figure 3a). To test whether the generated codes reserved the comparable cancer and tissue information, we adopted the five K-fold cross-validation (K = 5) technique to obtain unbiased generated samples and tested them on the trained cancer classifier and tissue classifier. The performance rates were 99.8 ± 0.2% and 95.0 ± 2.3% for the cancer type and tissue type, respectively. These nearly perfect classification performances suggest that the generated code has very similar cancer and tissue features to the true MET500 code, or the classifiers would not be able to classify them correctly.

To better show how MetGen reveals metastatic cancer distribution, we projected the true MET500 sample to generate the metastatic samples’ t-SNE plot (Figure 3a). We found that the original data points and generated data points had significant overlaps. To be more specific, for each generated cluster, there are original metastatic samples enclosed. That is to say, MetGen generated new samples around the original metastatic target samples and imputed space in between them. The more data are generated, the better the distribution pattern would show in the t-SNE plot. To quantitatively justify the results, we trained a classifier on our testing dataset to predict metastatic subtypes. The idea was to see whether the learned knowledge from the generated metastatic data can be applied to the true metastatic samples. Unsurprisingly, the model showed good predicting power for the true MET500 samples with an ROC AUC score of 97.6%, which further demonstrated MetGen’s power in generating high-quality metastatic samples from primary cancer and normal tissues.

To investigate one metastatic subtype specifically, we used breast cancer in liver as an example. We first retrieved all samples of breast cancer in liver from MET500, breast cancer samples, and normal liver tissue samples from TCGA, and then compared them with 1000 generated samples of breast cancer in liver in gene expression heatmaps (Figure 3b). The generated samples did not only have similar discriminative power but also looked like true samples in the gene expressions. Some genes were not active in primary breast cancer but were found to have high expression levels in metastatic breast cancer. Those genes were also upregulated in liver tissues. This implies that MetGen learns metastatic cancer by merging primary cancer information into the tissue environment.

### 3.3. Benchmark Using Stand-Alone MetVAE

In this section, we present a benchmarking analysis to evaluate the efficacy of the proposed MetGen framework against a generative model, specifically the Variational Autoencoder (VAE). Given that our MetVAE module embodies a VAE architecture, we leveraged its capabilities to directly generate samples and compare them with those produced by MetGen.

VAEs offer multiple approaches to sample generation: posterior sampling, prior sampling, and class-center sampling. While posterior sampling yields realistic data with limited variability, prior sampling produces highly variable but not always plausible data. Given our objective of generating specific metastatic subtypes, we employed class-center sampling for our benchmarking. This entails extracting the mean and standard deviation vectors from the latent nodes of the MET500 samples in MetVAE, followed by the computation of mean encodings for each subtype. Subsequent sample generation involves drawing from a normal distribution parameterized by these mean encodings.

We generated 200 samples for each metastatic subtype, totaling 3800 samples, using MetVAE. To assess the fidelity of the generated samples in preserving cancer and tissue information, we evaluated them using previously trained cancer and tissue classifiers. The classifiers achieved performance metrics of 78.9% and 65.4% for cancer and tissue type predictions, respectively. These results lag behind the performance of MetGen.

A deeper examination of the confusion matrices revealed that the subtypes BRCA, PRAD, and SARC exhibited relatively higher prediction accuracies, which was attributable to their larger training sample sizes facilitating more accurate distribution parameter learning in MetVAE (Appendix A). Notably, for tissue type prediction (Appendix A), only the liver samples were correctly classified, which is consistent with the observations in Figure 2c,d.

We attributed the suboptimal performance primarily to the insufficiency of samples and the lack of coherent clusters in the MetVAE latent space. This underscores the pivotal role of the contrastive framework employed by MetGen for this specific task.

### 3.4. MetGen Model Learns Metastatic Prostate Cancer Characteristics

The development of metastatic cancer requires cancer cells to leave their primary sites and eventually acclimate to new cellular surroundings in tissue sites. It is rather complicated than simply ‘put tumor to target location’. We used metastatic prostate cancer as an example to illustrate how MetGen takes advantage of its architecture to capture metastatic cancer information.

We generated 4000 metastatic prostate samples in four different tissue sites (lung, liver, bladder, and skin) using TCGA-PRAD and TCGA normal samples. Another 4000 samples were generated similarly, except the MetGen cancer branch was muted, and the generated samples did not contain prostate cancer information, namely the masked cancer samples (Figure 4a). To robustly study the characteristics of the samples, we used a gene set variation analysis (GSVA) [17] to perform a single-sample gene set analysis on 186 KEGG and 50 HALLMARK gene sets downloaded from the MSigDB collection [22,23]. The GSVA transformed the gene expression files into 236 gene set signatures. Eventually, we obtained a GSVA score matrix of 8000 by 236. Each number is a sample’s enrichment score for one specific gene set (Figure 4b). The metastatic prostate cancer and masked cancer metastatic samples clearly show different patterns. We anticipated that the difference is mainly attributed to the cancer information in metastatic prostate cancer.

We performed a differential analysis to identify the difference between the two groups (Figure 4c). The KEGG androgen response (AR), a key pathway in prostate cancer, ranked high in upregulation pathways. Multiple studies identified it in the development of the normal prostate and in the progression from primary to metastasis, and low AR transcriptional activity resulted in upregulated AR expression [24,25,26]. Some other well-known metastatic prostate pathways, such as purine metabolism [27,28,29] and complement and coagulation cascades [30,31], were also successfully detected by our model. Cancer intrinsic pathways like P53 signaling, along with WNT β catenin signaling, were found to be more active in the masked cancer samples, which also makes sense since prostate cancers are usually suppressive [32,33]. Furthermore, tumor microenvironment (TME) pathways, ECM receptor interaction [34,35], and epithelial mesenchymal transition [36,37] were found to be significantly differentially expressed in our comparison. The metastatic cascade is dependent on the loss of adhesion between cells, which results in the dissociation of the cell from the primary tumor and, subsequently, the ability of the cell to attain a motile phenotype via changes in the cell-to-matrix interaction (Figure 4c, Appendix A).

MetGen not only ranked prostate cancer-related pathways highly, but also correctly detected multiple cancer-intrinsic and metastatic TME pathways (Figure 4c). Such results prove that our model could leverage cancer information from the primary tumor and then embed it to the local tumor microenvironment.

### 3.5. MetGen Latent Components Learn Functional Clusters

The latent code learned from MetVAE reserves key metastatic cancer information; however, due to a limited number of metastatic cancer samples, statistically interpreting the code is often difficult. We used metastatic breast cancer in the bladder to illustrate how our model helps in latent functional interpretation.

We generated 1000 latent codes of metastatic breast cancer in the bladder using TCGA-BRCA and TCGA normal bladder tissues. Then, the hierarchical cluster was used to characterize 100 code components into clusters, and high concordance was shown among three functional clusters (Figure 5a).

To discover the pathway enriched in these functional clusters, we used a similar masking approach from a previous section. For instance, to obtain cluster 1 pathways, we set the cluster 1code components to zero, kept the rest of the components’ values as they were, and then fed the masked code to the MetVAE decoder to obtain masked metastatic cancer samples. Later on, the masked metastatic cancer samples will be compared to non-masked metastatic cancer samples; therefore, the difference is mainly attributed to cluster 1 components (Figure 5b).

Similar to the metastatic prostate cancer analysis, we first transformed our samples into KEGG and HALLMARK gene set signatures using GSVA. Since we had three masked cluster samples and non-masked samples, a 4000 by 236 GSVA score matrix was obtained. Figure 5c shows the heatmap of the activation level for gene sets in different samples.

To identify the cluster information, three differential comparisons were performed using LIMMA, for instance, metastatic breast cancer samples vs. each masked cluster sample. The top 20 positive and negative differential pathways from each comparison were selected for analysis (Appendix A). From the results (Figure 5d), we noticed that functional clusters often represent multiple functions. We roughly grouped them into five signatures.

#### 3.5.1. Cell Cycle

All three clusters detected that the G2M checkpoint, E2F targets, and MYC target V1 were upregulated. Those pathways are closely related to the cell cycle process, which is commonly active in biological samples [38,39,40,41].

#### 3.5.2. Immune Response

Immune response-related pathways are mostly located in cluster 2 and 3 components. In this group, we found an interferon alpha response, phosphatidylinositol signaling system, and PIK3, AKT, and mTOR signaling, which are actively regulated in immune activation and cell migration in the innate immune system [42,43]. Additionally, estrogen response was proven with an immunoenhancing effect on the immune system [44]. Together, the above pathways have been shown to regulate immune response by impairing the negative selection of high-affinity auto-reactive B cells, modulating B cell function, and leading to the Th2 response [45].

#### 3.5.3. Inflammatory/Cell Differentiation/Metastasis

This group has a rather homogeneous pathway score pattern but a very complicated signature. Clusters 1 and 3 detected both Hedgehog and Notch signaling pathways. Hedgehog is a signaling pathway that transmits information to embryonic cells required for proper cell differentiation. Notch signaling promotes proliferative signaling during neurogenesis, and its activity is inhibited by Numb to promote neural differentiation. They play a major role in the regulation of embryonic development [46,47,48]. Interestingly, clusters 1 and 3 also detected olfactory signaling and melanogenesis pathways, and along with the Hedgehog and Notch signaling pathways, are often found in melanoma tumor metastasis [49]. We also found several pathways related to specifical T cell response and immune cell trafficking in clusters 1 and 3. Antigen processing and presentation is a well-known pathway which can digest the proteins it encounters and displays peptide fragments from them on its surface for another immune cell to recognize [50,51]. The cytosolic DNA sensing pathway is responsible for detecting foreign DNA from invading microbes or host cells and generating innate immune responses [52]. Besides those pattern recognition receptors, immune cells such as the T cell receptor and NK-cell-mediated cytotoxicity signals are also activated, which strongly implies that this group is significant for inflammatory responses.

#### 3.5.4. Cellular Metabolism

Cellular metabolism involves complex sequences of controlled biochemical reactions, better known as metabolic pathways. These processes allow organisms to grow and reproduce, maintain their structures, and respond to environmental changes. We found two important cell metabolism target files, namely lipid metabolism and cytochrome P450 (CYP450). CYP450 is usually membrane-bound and localized to the inner mitochondrial or endoplasmic reticular membrane [53]. The lipid metabolism is an important source of cellular energy, which is stored as triglycerides [54]. Both play critical roles in oxygenase activity [55].

#### 3.5.5. Bladder Function

Cluster 2 detected a few non-cancer-related pathways, such as vascular smooth muscle contraction, tight junction, and calcium signaling pathways [56,57]. Those pathways are highly correlated with muscle activities of the bladder. We believe metastatic tissue site information is mainly stored in this cluster.

The above results show that MetGen latent code components allow for great interpretability. Each functional code cluster captured multiple functional processes, which are highly correlated with metastatic breast cancer. The generated samples reveal the characteristics of metastasis, target tissue sites, and regular cancer-related immune responses.

## 4. Conclusions

We proposed an interpretable deep-learning framework inspired by contrastive learning for metastatic cancer sample generation using primary tumor and normal tissue samples. The pipeline consists of two model components, MetVAE and MetGen. We comprehensively investigated our models’ performance and interpretability using TCGA and MET500 data. We showed the following:MetVAE can encode metastatic cancer and tissue site information faithfully into latent code. We investigated MetVAE for cancer and tissue type classification using MET500 data, and our model had good performance in both tasks.MetGen can generate metastatic cancer expressions from primary tumor tissues and normal tissues. We generated 19 metastatic cancer types using TCGA data. Our generated samples stored essential metastatic cancer information and achieved good performance in multiple classification tests.We demonstrated the interpretability of our models using samples of metastatic prostate cancer and metastatic breast cancer in the bladder. Highly relevant functions were learned from primary cancer and tissue sites, which further affirmed the power of our model.

Inevitably, there are still many challenges to be addressed. Metastatic tumors are highly microenvironment-dependent; right now, our model only provides a general environment tissue site. For the same reason, our model cannot deal with scenarios such as when a cancer cell travels from the left breast to the right breast. Research investigating solutions to these challenges will be our future focus.

## Figures and Tables

**Figure 1 cancers-16-01653-f001:**
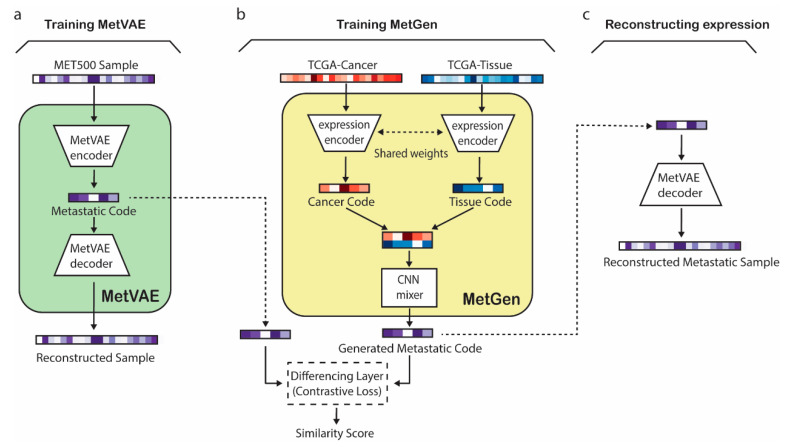
Overview of MetGen workflow. Solid arrows indicate model training path, while dashed arrows indicate workflow path. (**a**) MetVAE model training illustration. (**b**) MetGen model learns metastatic code using contrastive learning. (**c**) Metastatic sample reconstruction using MetVAE decoder.

**Figure 2 cancers-16-01653-f002:**
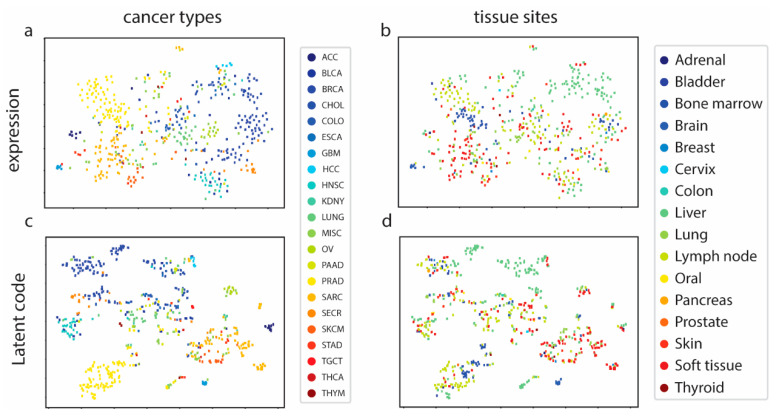
MetVAE learns cancer and tissue patterns. (**a**,**b**) MET500 samples in expression space colored by cancer types and tissue sites. (**c**,**d**) MET500 samples in latent code space colored by cancer types and tissue sites.

**Figure 3 cancers-16-01653-f003:**
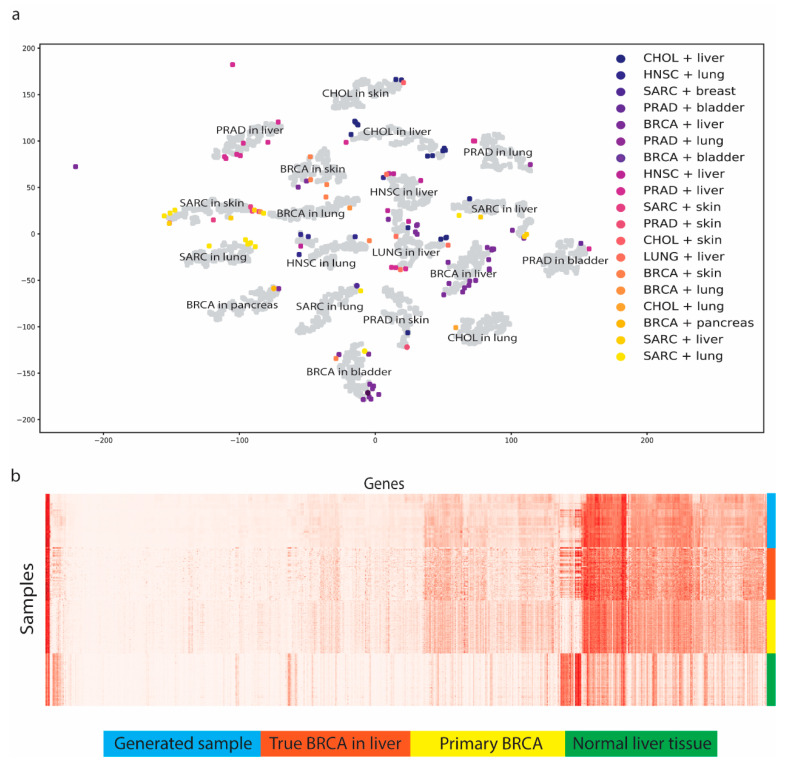
MetGen generates metastatic cancer samples. (**a**) t-SNE plot of generated metastatic samples (grey) represent MET500 samples (colored). Metastatic types of generated samples are marked in text. (**b**) Expression of generated BRCA in liver samples compared to true BRCA in liver, primary BRCA, and normal liver tissue.

**Figure 4 cancers-16-01653-f004:**
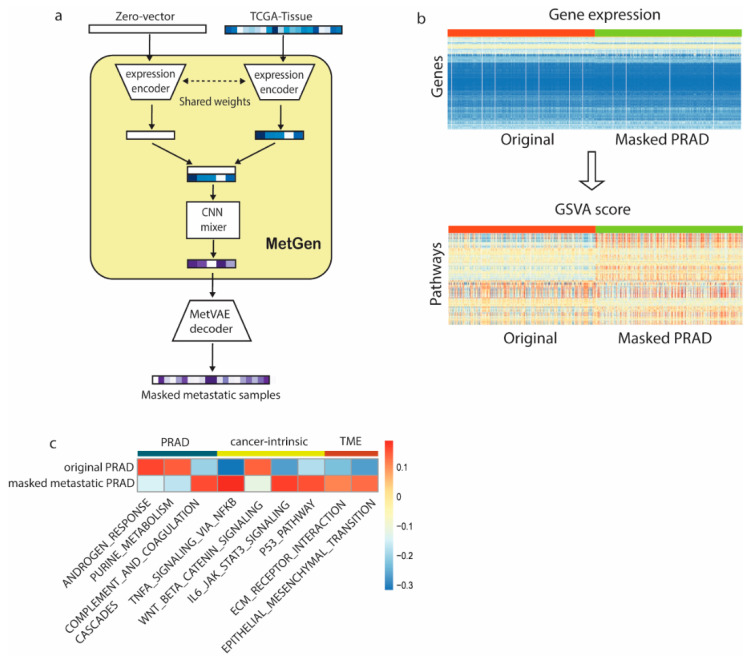
MetGen model learns metastatic prostate cancer characteristics. (**a**) Workflow of metastatic prostate cancer study. (**b**) Converting gene expression profiles to pathway expression; red color represents original generated samples, and green represents masked samples. (**c**) Metastatic cancer-related differential pathways.

**Figure 5 cancers-16-01653-f005:**
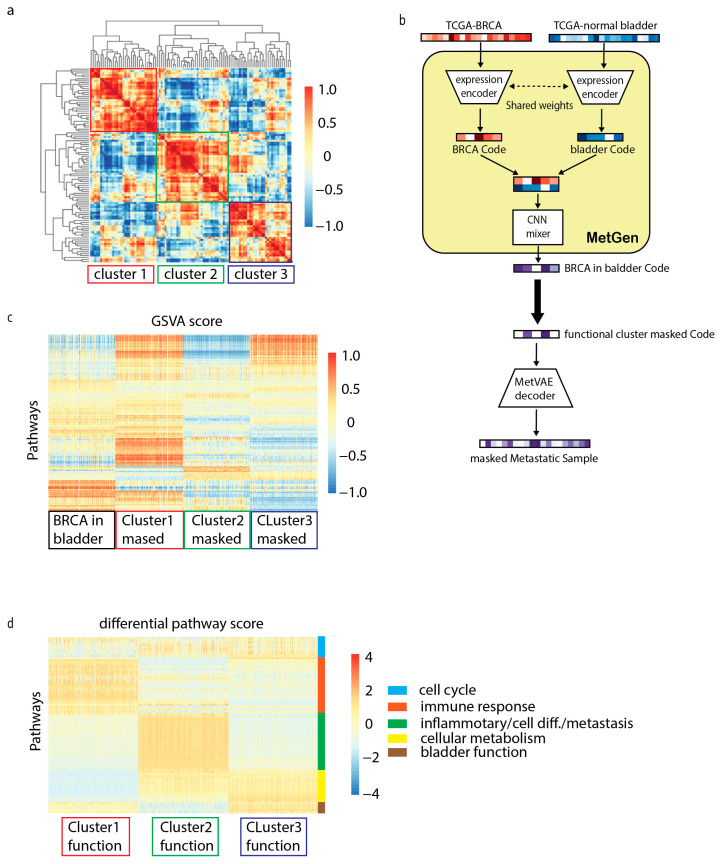
MetGen learns functional components in metastatic breast cancer in bladder. (**a**) Hierarchical clustermap of generated codes for metastatic breast cancer in bladder. (**b**) Analysis workflow of functional components. (**c**) Heatmap of GSEA score for 4 group samples. (**d**) Selected differential pathways. Padj < 0.05 was used for significant cutoff. Top 10 up or downregulated pathways in fold change are selected for analysis.

**Table 1 cancers-16-01653-t001:** Data table.

TCGA Cancer Type	Number of Samples	TCGA Normal Tissue Sites	Number of Samples
BRCA	159	bladder	8
CHOL	45	breast	5
HNSC	45	liver	243
LUNG	52	lung	75
PRAD	155	pancreas	1
SARC	100	skin	40

**Table 2 cancers-16-01653-t002:** MetVAE encoder structures.

Layer (Type)	Output Shape	Number of Param	Connected to
encoder_input	(None, 7312, 1)	0	
Flatten_1	(None, 7312)	0	encoder_input
dense	(None, 1000)	7,313,000	Flatten_1
batch_normalization	(None, 1000)	4000	Dense
Flatten_2	(None, 1000)	0	batch_normalization
z_mean	(None, 100)	100,100	Flatten_2
z_log_var	(None, 100)	100,100	Flatten_2
z_sample	(None, 100)	0	z_mean, z_log_var

**Table 3 cancers-16-01653-t003:** MetVAE decoder structures.

Layer (Type)	Output Shape	Number of Param
z_sampleing(input)	(None, 100)	0
batch_normalization	(None, 100)	400
dense	(None, 1000)	101,000
batch_normalization	(None, 1000)	4000
dense	(None, 7312)	7,319,312
reshape	(None, 7312, 1)	0

**Table 4 cancers-16-01653-t004:** TCGA encoder structures.

Layer (Type)	Output Shape	Number of Param
TCGA_input	(None, 7312, 1)	0
conv1d	(None, 228, 64)	2112
batch_normalization	(None, 228, 64)	256
activation	(None, 228, 64)	0
dense	(None, 57, 16)	4112
batch_normalization	(None, 57, 16)	64
activation	(None, 57, 16)	0
flatten	(None, 912)	0
dropout	(None, 912)	0
dense	(None, 912)	467,456

**Table 5 cancers-16-01653-t005:** CNN mixer structures.

Layer (Type)	Output Shape	Number of Param	Connected to
Cancer_input	(None, 512)	0	
Tissue_input	(None, 512)	0	
stack	(None, 2, 512, 1)	0	Cancer_input, Tissue_input
Conv2d_1	(None, 1, 512, 64)	192	stack
batch_normalization_1	(None, 1, 512, 64)	256	Conv2d_1
Conv2d_2	(None, 1, 16, 32)	65,568	batch_normalization_1
batch_normalization_2	(None, 1, 16, 32)	128	Conv2d_2
flatten	(None, 512)	0	batch_normalization_2
Dropout_1	(None, 512)	0	flatten
dense	(None, 200)	102,600	Dropout_1
Dropout_2	(None, 200)	0	dense
MET500_code(input)	(None, 100, 1)	0	
Learned_code	(None, 100)	20,100	Dropout_2

**Table 6 cancers-16-01653-t006:** Standard DNN structures.

Layer (Type)	Output Shape	Number of Param
embeddings (input)	(None, 100)	0
dense	(None, 60)	6060
dropout	(None, 60)	0
dense	(None, 40)	2440
dropout	(None, 40)	0
dense	(None, number of classes)	
classifier	(None, number of classes)	

## Data Availability

All data used in this study are publicly available on XenaHub; the code is available at https://github.com/Fclef/MetGen (accessed on 9 April 2024).

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
