# Peer review of "A Metastatic Cancer Expression Generator (MetGen): A Generative Contrastive Learning Framework for Metastatic Cancer Generation"

_cancers, 2024, doi:10.3390/cancers16091653_

Round 1
Reviewer 1 Report
Comments and Suggestions for Authors
This paper presents a technique titled, “Metastatic cancer expression generator (MetGen): A generative contrastive learning framework for metastatic cancer generation”. The topic is good however I have some points that need to be addressed given as under.
My detailed concerns on the study are:
1. The abstract must be rewritten to show some benefits and write what is new for the current state of knowledge.
2. In the abstract add some quantitative values of the results.
3. The introduction section may be split into two sections Introduction and literature review.
4. In the introduction section clearly indicate the contributions of the current study.
5. For more clarity, the parameters used to implement the metastatic cancer expression generator (MetGen) model may be presented in tables.
6. Statistical analysis of the proposed model may be added to show the prominence of the proposed model.
7. Figure captions should be written under the figures.
8. Indicate the adding number to the sub-headings in each section.
9. The Methodology section-3 must be written before the Results and Discussion section-2.
10. I could not find the comparison of the proposed method with the state-of-the art that may be added in the experiments section.
11. Add some statistical analysis to exhibit the prominence of the proposed work.
Reviewer 2 Report
Comments and Suggestions for Authors
The manuscript focuses on leveraging deep learning techniques and a contrastive learning approach to generate expressions related to metastatic cancer. Utilizing a combination of data from TCGA and MET500 datasets, positive and negative pairs were curated for training and inference. The MetVAE model was employed to encode latent information from the MET500 dataset, while the MetGen framework facilitated the generation of metastatic cancer codes through contrastive learning. The generated dataset underwent classification tests to evaluate its efficacy. Additionally, the manuscript conducts tests such as Gene Set Variation Analysis and Differential Analysis, providing insights into pathway expression and enhancing understanding of the underlying mechanisms.
Few questions on the manuscript,
1. The testing dataset was generated from the same pool of training data. The testing dataset of 200 pairs per metastatic subtypes were created from the same pool as the training and validation dataset. This raises a concern of data being generated very similar or same as the training dataset.
2. The results look promising for the generated models. Was there a comparison done between the results from other frameworks against the contrastive learning MetGen framework.
3. Testing across the (3800) generated dataset, and to check the generalization of the model, a cross-validation study and results would be better to understand the generalizability of the model performance. The cross-validation study was not provided in the current manuscript.
4. The classifier used to distinguish the MET500 codes and to predict cancer types and tissue types was not explained, and the model architecture (diagram) was not available in the manuscript.
5. The pre-trained model was used to classify the generated test dataset into cancer type and tissue type, and the performance was about 99.9% and 96.6%. The dataset used for training and testing the data was collected from the same TCGA cancer, TCGA Tissue and MET500 dataset. It would be interesting to check if there was any data leakage between the training and test set as they booth are originating from the same parent dataset.
6. Also, the model architecture used for classifying the metastatic subtypes was not provided in the manuscript.
Comments on the Quality of English LanguageEnglish language requires minor improvement.
Round 2
Reviewer 1 Report
Comments and Suggestions for Authors
The authors have responded to all my comments, I have no more comments and the paper may be considered for publication.